# Genotype-Phenotype Correlation in Hypertrophic Cardiomyopathy: New Variant p.Arg652Lys in *MYH7*

**DOI:** 10.3390/genes13020320

**Published:** 2022-02-09

**Authors:** Guido Antoniutti, Fiama Giuliana Caimi-Martinez, Jorge Álvarez-Rubio, Paula Morlanes-Gracia, Jaume Pons-Llinares, Blanca Rodríguez-Picón, Elena Fortuny-Frau, Laura Torres-Juan, Damian Heine-Suner, Tomas Ripoll-Vera

**Affiliations:** 1Cardiology Department, Hospital Universitario Son Llàtzer, 07198 Palma de Mallorca, Spain; guidoantoniutti@hotmail.com (G.A.); fiamacaimi91@gmail.com (F.G.C.-M.); jalvarezr@hsll.es (J.Á.-R.); 2Health Research Institute of the Balearic Islands (IdISBa), 07120 Palma de Mallorca, Spain; jaumea.pons@ssib.es (J.P.-L.); elena.fortuny@ssib.es (E.F.-F.); laura.torresjuan@ssib.es (L.T.-J.); damian.heine@ssib.es (D.H.-S.); 3Cardiology Department, Hospital Clínico Universitario Lozano Blesa, 50009 Zaragoza, Spain; pmorlanesg@gmail.com; 4Cardiology Department, Hospital Universitario Son Espases, 07120 Palma de Mallorca, Spain; 5Cardiology Unit, Hospital Mateu Orfila, 07703 Menorca, Spain; shirljovi@gmail.com; 6Unit of Molecular Diagnostics and Clinical Genetics, Hospital Universitario Son Espases, 07120 Palma de Mallorca, Spain; 7CIBEROBN (Physiopathology of Obesity and Nutrition), 28029 Madrid, Spain

**Keywords:** hypertrophic cardiomyopathy, cardiomyopathies, cardiomyopathy, genetic, genetic testing, next-generation sequencing, NGS for diagnostics of CVDs, variant interpretation, variant classification

## Abstract

Hypertrophic cardiomyopathy (HCM) is a genetic disease characterised by increased left ventricle (LV) wall thickness caused by mutations in sarcomeric genes. Finding a causal mutation can help to better assess the proband’s risk, as it allows the presence of the mutation to be evaluated in relatives and the follow-up to be focused on carriers. We performed an observational study of patients with HCM due to the novel p.Arg652Lys variant in the *MYH7* gene. Eight families and 59 patients are described in the follow-up for a median of 63 months, among whom 39 (66%) carry the variant. Twenty-five (64%) of carriers developed HCM. A median maximum LV wall thickness of 16.5 mm was described. The LV hypertrophy was asymmetric septal in 75% of cases, with LV outflow tract obstruction in 28%. The incidence of a composite of serious adverse cardiovascular events (sudden death, aborted sudden death, appropriate implantable cardiac defibrillator discharge, an embolic event, or admission for heart failure) was observed in five (20%) patients. Given the finding of the p.Arg652Lys variant in patients with HCM, but not in controls, with evident segregation in patients with HCM from eight families and the location in an active site of the protein, we can define this variant as likely pathogenic and associated with the development of HCM.

## 1. Introduction

Hypertrophic Cardiomyopathy (HCM) is a genetic disease characterised by increased left ventricle (LV) wall thickness caused by variants in sarcomeric genes [1]. To date, more than 1500 variants associated with the disease have been described, usually with an autosomal dominant pattern of inheritance. Each child of a patient with the variant has a 50% chance of inheriting it, although penetrance is incomplete and the age of presentation in those who develop the disease is variable [2]. The most frequently affected genes are *MYH7* and *MYBPC3*, which encode the β-myosin heavy chain (β-MHC) and the myosin binding protein C, respectively. Together, variants in *MYH7* and *MYBPC3* are identified in 70% of patients with HCM and a positive result in the genetic analysis [3].

Although the hereditary nature of the disease has been known since 1961 [4], *MYH7* was the first gene described to cause HCM in 1990 [5,6]; since then, multiple variants capable of producing the disease have been discovered [7,8].

Finding a causal variant of the disease in patients with HCM can sometimes help to better assess the proband’s risk; however, above all, it allows the presence of the variant to be evaluated in relatives and the follow-up to be focused on carriers [9]. The profitability of the genetic study in patients affected by HCM is around 50%, despite the new technologies that are available [10]. Describing new pathogenic or likely pathogenic variants increases diagnostic performance, allows better risk characterization, reduces the number of patients in follow-up after ruling out their presence, and reduces health costs [11].

We have observed the presence of a new variant at position 1955 of *MYH7*, which causes the substitution of the amino acid arginine at position 652 with lysine (NP_000248.2:p.Arg652Lys, NM_000257.2:c.1955G>A) in β-MHC in a series of patients diagnosed with HCM belonging to non-related families in a community in Spain. This finding suggests that the p.Arg652Lys variant could be associated with the development of the disease.

## 2. Objective

To determine the genotype-phenotype association of the new variant p.Arg652Lys in *MYH7* in patients with HCM.

## 3. Materials and Methods

A retrospective and observational study in which patients with a diagnosis of HCM and a finding of the p.Arg652Lys variant in *MYH7* were analysed. From those patients, family screening was carried out, drawing up a genealogical tree of at least three generations and studying the first-degree relatives of the affected individuals both clinically and genetically.

### 3.1. Patient Selection

Patients that followed-up in the cardiology services of the Balearic Islands (Spain) with a diagnosis of HCM were analysed using phenotypic and genetic investigations between January 2001 and April 2021. The patients with a finding of the p.Arg652Lys variant were selected.

The family histories of at least three generations of each proband were investigated (Appendix A), and the phenotype of the first-degree relatives of those who presented the variant are described (Figure 1).

HCM was defined as the presence of a maximum left ventricular wall thickness (LVWT) equal to or greater than 15 mm, determined by cardiac imaging techniques, or equal to or greater than 13 mm in those with a family history of HCM, according to the criteria expressed in the latest consensus from the European Society of Cardiology in 2014 [12,13] and the American College of Cardiology/American Heart Association in 2020.

### 3.2. Genetic Analysis

Given the diagnosis of HCM in the index cases, a genetic analysis was carried out to determine the presence of variants associated with the disease. The samples were analysed in three different Spanish laboratories: Health in Code (A Coruña, Spain), Imegen (Valencia, Spain), and Unidad de Diagnóstico Molecular y Genética Clínica del Hospital Universitario Son Espases (Mallorca, Spain), according to availability at the time of evaluation of each patient.

The technique used for index patients was Sanger or Next Generation Sequencing (NGS). A minimum of five genes (Sanger) and a maximum of 287 genes (NGS) were analysed, among which the five main sarcomeric genes were always included (*MYH7*, *MYBPC3*, *TNNT2*, *TNNI3*, and *TPM1*).

Variants were filtered using a pre-established protocol based mainly on the probable functional impact on the protein and allelic frequency. Predictive bioinformatics tools were used “in silico”. We applied the 2015 consensus guidelines of the American College of Medical Genetics and Genomics and the Association for Molecular Pathology (ACMG/AMP) [14] to classify the variants as pathogenic, likely pathogenic, or variants of uncertain significance. We excluded those considered benign, probably benign, and variants of uncertain significance with a frequency rate ≥ 0.02% in the GnomAD databases and our private database at Unidad de Diagnóstico Molecular y Genética Clínica del Hospital Universitario Son Espases or in those where co-segregation could not be demonstrated in the cases studied.

The genetic cascade study of the relatives was aimed at evaluating exclusively the variants present in the index case.

### 3.3. Clinical Evaluation and Complementary Studies

Clinical evaluation consisted of the preparation of a detailed medical history, the constitution of a family tree, and a complete physical examination performed by cardiologists belonging to the Family Heart Disease Unit of the Son Llàtzer University Hospital (Palma de Mallorca), Son Espases University Hospital (Palma de Mallorca), and Hospital Mateu Orfila (Mahón, Menorca).

The complementary studies included an electrocardiogram (ECG) and a transthoracic echocardiogram (TTE) both in the index cases and in the family screening study. Cardiac Magnetic Resonance (MRI), 24-h Holter ECG, and stress tests were performed according to the discretion of the treating physician. All investigations were carried out by cardiologists and verified by specialists in family heart disease.

After the clinical evaluation, the risk of sudden death at 5 years in patients with HCM was analysed using the risk calculator proposed by O’Mahony et al. in 2014 [15]. The classifications are as follows: low risk for those who presented a value of <4%, moderate risk for those who presented a value between 4 and 6%, and high risk for those who presented a value that exceeded 6%.

### 3.4. Statistical Analysis

A descriptive analysis of all variables was carried out. With the categorical variables, the frequencies and global percentages were estimated. Normality tests and graphs were used to determine whether the quantitative variables followed a normal distribution. Variables with normal distribution were expressed as mean ± standard deviation and those that did not have a normal distribution as a median and [interquartile range]. To describe the significance and associations between two variables, the chi-square test was used to face two qualitative categorical variables, and the Mann–Whitney U test was used to face a non-parametric quantitative variable to a qualitative one. A value of *p* < 0.05 was considered to be an indicator of a significant difference. SPSS v.23 software was used for data analysis.

## 4. Results

### 4.1. Description of the Analysed Population

Eight families are described in the follow-up, with at least one patient diagnosed with HCM in the presence of the p.Arg652Lys variant (Table 1). A screening study was carried out in relatives of patients with HCM and with the p.Arg652Lys variant, which provided a total of 59 patients; of those, 39 (66%) presented the variant in a heterozygous state.

Among the patients with the p.Arg652Lys variant, follow-up was performed for a median of 63 months (IQR (25-75): 23–112), observing the development of heart disease in 25 (64%) carriers, with HCM being the only observed phenotype. No development of heart disease was detected in patients who did not carry the p.Arg652Lys variant. The median age of the patients at the time of the first clinical contact was 39 years (IQR [25-75]: 30.7–57.5), with those who carried the variant and had not developed HCM being younger (mean age 25.1 vs. 51.4; *p* < 0.0001); that difference persisted in the last follow-up visit (mean age 29.6 vs. 58.2; *p* < 0.0001). The ethnicity of all patients was Caucasian. Overall, 49% of carriers and 46% of those who developed HCM were female (Table 2).

In the group of patients who developed HCM, a median Maximum LVWT of 16.5 mm (IQR [25-75] 13.8–22 mm) was described at follow-up. The morphology of ventricular hypertrophy was asymmetric septal in 75% of patients, concentric in 17% of patients, and apical in 8% of patients. A total of 28% of the patients presented obstruction to the LV outflow tract (>30 mmHg) during follow-up, 8% presented episodes of non-sustained ventricular tachycardia (NSVT) using a 24-h Holter ECG, 4% presented associated right ventricular hypertrophy, 12% had atrial fibrillation (AF), and none reported impaired left ventricular function.

In 75% of the genetic studies of the index patients, the p.Arg652Lys variant was found in isolation, while a second variant associated with heart disease was found in two (25%) of them.

In the first case, the p.Arg652Lys variant in *MYH7* and the p.Arg190Trp variant in KCNQ1 were found. The latter is classified as likely pathogenic and associated with Long QT Syndrome, but no association with HCM has been found. In the second case (Figure 2), in addition to the p.Arg652Lys variant, an intronic variant of uncertain significance was found in the *TPM1* gene (c.375-5T>C), which does not segregate the disease in the family.

### 4.2. Analysis of Events during the Follow-Up

In order to determine the risk of an unfavorable clinical course, the incidence of a composite of serious adverse cardiovascular events (ACEs) in patients with HCM, including sudden death (SD), aborted sudden death (ASD), appropriate implantable cardiac defibrillator (ICD) discharge, embolic events (stroke or peripheral ischemic event), and admission for heart failure (HF) in patients diagnosed with HCM was analysed.

The ACE compound was observed in five patients (20%) (Table 3). The individual analysis of the variables revealed one (4%) patient with SD, one (4%) with ASD, and three (12%) with embolic events, without registering any appropriate ICD discharge or admission for HF. Except for the ASD event that occurred at the age of 48, all other ACEs occurred at an age of 64 or higher (ACE median age 69.5; IQR [25-75]:60–75.5).

The patients who presented ACEs had a higher incidence of AF (40% vs. 5%, *p* = 0.031).

The patients who presented SD and ASD at the time of the event had a low risk of sudden death (1.25% and 2.11%, respectively), according to the calculator by O’Mahony et al., and a mean score that did not differ significantly from that calculated in the group that did not present SD or ASD (1.68 ± 0.41 vs. 2.33 ± 0.31, *p* = 0.53).

It should be noted that we observed an additional SD event in a 70-year-old male patient, a carrier of the genetic variant p.Arg652Lys, who was genetically investigated after an ASD event in his sister. Since he did not have any cardiological follow-up, it is not possible to determine his clinical condition or to consider it to be sudden death associated with HCM in the event analysis.

Regarding invasive treatment, the need for septal reduction therapy has been described in one (4%) patient due to the persistence of symptoms despite optimal medical treatment, with surgical myectomy being the treatment applied with good subsequent clinical evolution. In three (12%) patients, the implantation of an ICD was indicated with the objective of primary prevention of sudden death, with the ICD being rejected by one of them and with the implantation being completed in the rest. No patients with LVEF deterioration < 50% at the time of diagnosis or during follow-up were found, and neither did patients with advanced heart failure require the transplantation or implantation of ventricular assist devices.

## 5. Discussion

Patients with HCM carry an increased risk of ACE throughout life. HCM has a prevalence of between 1/200 and 1/500 people [16] and is the leading cause of SD in patients under 35 years of age [17].

The genetic variant p.Arg652Lys is a missense change in the *MYH7* gene that has not been described in population genetic studies. The GnomAd database (v2.1.1 data set that spans 125,748 exome sequences and 15,708 whole-genome sequences), ClinVar, and our private database at Unidad de Diagnóstico Molecular y Genética Clínica del Hospital Universitario Son Espases (including 1017 exomes from patients with very diverse diseases) were searched, and no description of the p.Arg652Lys variant was found. No publications consistent with its presence were associated with the development of HCM or any other form of heart disease.

In 2002, MacCoss et al. [18] identified the p.Arg652 position of β-MHC as an important methylation site. However, the physiological significance of this modification is not conclusive.

There are missense-like variants that affect amino acids close to position 652 in β-MHC, which have been associated with the development of HCM.

The substitution of arginine for lysine is conservative in terms of the degree of structural variation of the molecule because both amino acids are similar, as they are basic residues. Despite this, position p.Arg652 is highly conserved in β-MHC, so a variant in this position could be detrimental.

A variant at this site that causes the amino acid substitution from arginine to glycine (p.Arg652Gly) has been associated with the development of the disease [19,20] and is considered pathogenic in the development of HCM [21]. Although the variant described in our population shares the site of involvement, we cannot guarantee that it has the same functional effect.

A segregation analysis of the p.Arg652Lys variant was performed with the development of HCM in the aforementioned families, observing joint segregation of the variant and the HCM phenotype.

Following the ACMG/AMP classification (Table 4), we can see that the variant p.Arg652Lys meets the pathogenicity criteria based on the population information. This is because it is absent from population databases and segregation, after having described co-segregation in the eight families studied, and because it affects an active site of the protein β-MHC where a variant in the same amino acid has already been clearly established as pathogenic (p.Arg652Gly).

Considering the p.Arg652Lys variant as likely pathogenic and associated with the development of HCM, it is possible to analyse its clinical behaviour. Penetrance is incomplete, reaching 64% in our population, and expressiveness is variable, with a very wide range of clinical expression ranging from the sole presence of left ventricular hypertrophy with no other associated manifestation to sudden death events at an age below 50 years.

The risk assessment shows a behaviour of high clinical risk, with a high incidence of ACE that reaches 20% of the affected patients and 8% overall between SD and ASD events in patients with a confirmed HCM phenotype. Even though it is not possible to correctly describe an age-dependent analysis due to the small number of events, we can observe that adverse events occur at an advanced age, with only one ACE event at under 50 years of age.

Although we can observe that the HCM Risk-SCD score [15] does not differ significantly between the groups of patients who presented SD/ASD events and those who did not, it is not possible to define whether it is a useful risk stratification method in our study population since the small number of events makes statistical interpretation difficult.

## 6. Limitations

The present study is based on a small population of patients who have been followed for a limited period of time. Analysis in a larger numbers of patients is required over a long follow-up period to determine the true risk of adverse events in HCM patients due to the p.Arg652Lys variant.

## 7. Conclusions

Given the finding of the p.Arg652Lys variant in patients with HCM in our setting, but not described in controls, with evident segregation in patients with HCM from eight families and the location in an active site of the protein, which coincides with another variant previously described as pathogenic, we can define it as a likely pathogenic variant associated with the development of HCM, with incomplete penetrance and a high clinical risk in the series of patients described. The exclusive presence of the variant in our region could correspond to a founder effect in the Balearic Islands, Spain, which could be confirmed after further investigation.

## Figures and Tables

**Figure 1 genes-13-00320-f001:**
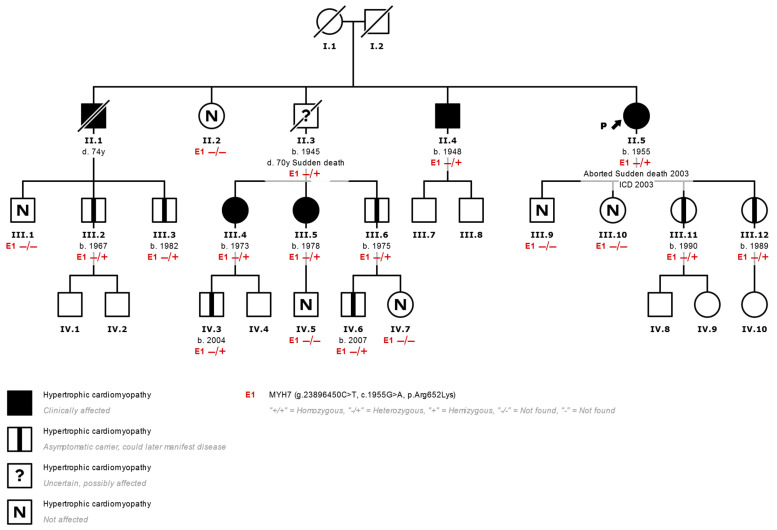
Family tree of a typical family with HCM. (II.5 corresponds to index case N°7 in Table 1) Most HCMs are inherited in an autosomal dominant fashion. Affected individuals are shown in black, and healthy carriers of the p.Arg652Lys variant (E1) are shown with a central black mark. Using a genetic panel, the variant was identified in the proband (arrow) after an episode of aborted sudden death. Other members of the family who inherited the variant were affected clinically but without severe events such as the proband, which shows variable expressivity of the variant. ICD: Implantable Cardiac Defibrillator. −/−: Variant not found. −/+: Heterozygous variant found.

**Figure 2 genes-13-00320-f002:**
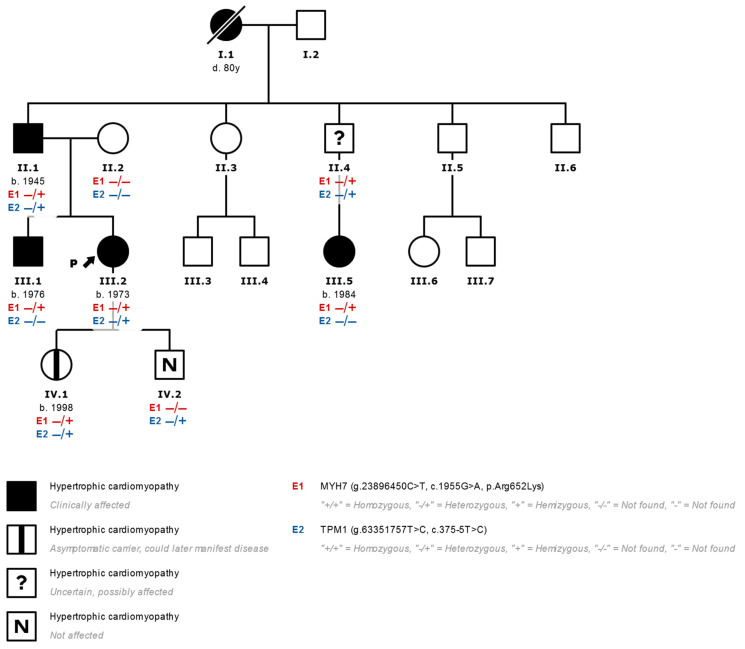
Family tree of a family studied with the finding of two variants that are potentially associated with heart disease (III.2 corresponds to index case N°5 in Table 1). In genetic studies of patients with HCM and the finding of more than one variant that is not clearly associated with HCM, the pathogenicity of all the variants found should be considered and their family segregation evaluated. In this case, the presence of variants in the *MYH7* (E1) and *TPM1* (E2) genes is seen in the index case (arrow) with a diagnosis of HCM (effects indicated in black). We observed that only the variant p.Arg652Lys in *MYH7* segregates together with the development of HCM in relatives, which is evident in the brother and in a cousin of the index case who presented HCM and only presented the variant in *MYH7* without alterations in *TPM1*. Patients with the *MYH7* variant and no HCM phenotype are marked with a central black mark. −/−: Variant not found. −/+: Heterozygous variant found.

**Table 1 genes-13-00320-t001:** Characteristics of the population with findings of the p.Arg652Lys variant in *MYH7*.

Population Characteristics															Clinical Behavior					
Family Number	Relation to Index Case	Sex	Age at First Medical Contact	Phenotype	Follow-Up Time (Months)	SD FH in 1st Degree Relatives	Clinical Presentation	AF	NYHA	MLVWT in 1st TTE (mm)	MLVWT in Follow-Up (mm)	LVOT Obstruction >30 mmHg	LVEF at Diagnosis (%)	LA Maximum Diameter in Follw-Up (mm)	NSVT	Septal Reduction Therapy	HF	Embolic Event (Age)	SD (Age)	ASD (Age)
1	Index case	M	34	HCM	127	No	Sincope	No	1	22	24	Yes	?	35						
	Sister	F	39	Healthy Carrier	114	No	Casual or family Screening	No	1	11	11	No	65	35						
2	Index case	F	79	HCM	9	No	Palpitations	Yes	1	15	15	No	60	43,7			Yes	Yes (70)		
	Son	M	51	HCM	113	No	Casual or family Screening	No	1	12	13	No	60	35						
	Son	M	45	HCM	129	No	Casual or family Screening	No	1	13	13	No	60	35						
	Grandson	M	15	Healthy Carrier	112	No	Casual or family Screening	No	1	9	9	?	?	?						
3	Index case	M	31	HCM	198	No	Angina	No	1	18	18	Yes	60	43		Yes				
	Mother	F	73	HCM	15	No	Angina	No	1	22	22	Yes	71	58						
4	Index case	F	53	HCM	197	No	Angina	No	1	11	18	No	?	35			Yes			
	Daughter	F	30	Healthy Carrier	63	No	Casual or family Screening	No	1	11	11	No	60	35						
	Aunt	F	80	HCM	114	No	Palpitations	Yes	1	12	12	No	71	55				Yes (92)		
5	Index case	F	40	HCM	62	No	Angina	No	1	20	21	Yes	60	51						
	Brother	M	38	HCM	74	No	Casual or family Screening	No	1	19	22	Si	60	39	Yes					
	Daughter	F	16	Healthy Carrier	85	No	Casual or family Screening	No	1	6	9	No	60	35						
	Father	M	70	HCM	17	No	Casual or family Screening	No	1	14	14	No	55	39				Yes (69)		
	Niece	F	31	HCM	15	No	Casual or family Screening	No	1	?	?	?	?	?						
	Uncle	M	65	HCM	69	No	Casual or family Screening	No	1	13	13	No	65	41						
6	Index case	M	45	HCM	207	No	Casual or family Screening	No	?	15	15	?	?	35					Yes (64)	
	Sister	F	56	HCM	0	Yes	Casual or family Screening	No	1	?	?	?	?	?						
	Brother	M		HCM	0	No	Casual or family Screening	Yes	?	?	?	?	?	?						
	Sister	F	61	HCM	20	Yes	Casual or family Screening	No	1	15	17	Yes	70	40						
	Daughter	F		Unknown	0	Yes	Casual or family Screening	No	?	?	?	?	?	?						
	Grandson	M	32	HCM	15	No	Casual or family Screening	No	1	16	16	?	62	35						
7	Index case	F	41	HCM	291	No	Dyspnoea	No	2	?	?	?	?	?						Yes (48)
	Brother	M		Unknown	0	Yes	Casual or family Screening	No	?	?	?	?	?	?					Yes (70)	
	Brother	M	69	HCM	35	Yes	Casual or family Screening	No	1	22	22	No	62	42	Yes					
	Daughter	F	25	Healthy Carrier	63	Yes	Casual or family Screening	No	1	9	9	No	65	33						
	Daughter	F	24	Healthy Carrier	64	Yes	Casual or family Screening	No	1	12	12	No	71	35						
	Niece	F	37	Healthy Carrier	55	Yes	Casual or family Screening	No	1	8	9	No	69	34						
	Nephew	M	39	HCM	65	Yes	Casual or family Screening	No	?	13	16	Yes	75	40						
	Niece	F	41	Healthy Carrier	0	Yes	Casual or family Screening	No	1	8	8	No	66	30						
	Niece	F	46	HCM	91	No	Casual or family Screening	No	1	13	22	Si	80	37						
	Nephew	M	30	HCM	92	No	Casual or family Screening	No	1	16	25	No	75	43						
	Grand nephew	M	11	Healthy Carrier	26	No	Casual or family Screening	No	1	6	6	No	74	26						
	Grand nephew	M	12	Healthy Carrier	43	No	Casual or family Screening	No	1	10	11	No	71	33						
8	Index case	F	59	HCM	68	Yes	Casual or family Screening	No	1	16	17	No	60	39						
	Son	M	19	Healthy Carrier	62	No	Casual or family Screening	No	1	9	10	No	67	37						
	Cousin	M	65	HCM	0	No	Casual or family Screening	No	1	13	13	No	63	39						
	Second nephew	M	32	Healthy Carrier	0	No	Casual or family Screening	No	1	9	9	No	68	33						

?: Unknown, ACE: Adverse Cardiovascular Event, AF: Atrial Fibrillation, ASD: Aborted Sudden Death, F: Female, FH: Family History, HCM: Hypertrophic Cardiomyopathy, ICD: Implantable Cardiac Defibrillator, LA: Left Atrium, LVEF: Left Ventricle Ejection Fraction, LVOT: Left Ventricle Outflow Tract, M: Male, MLVWT: Maximum Left Ventricle Wall Thickness, NSVT: Non-sustained Ventricular Tachycardia, NYHA: New York Heart Association dyspnea classification, PCM: Pacemaker, SD: Sudden Death, SDPP: Sudden Death Primary Prevention, SDSP: Sudden Death Secondary Prevention, TTE: Transthoracic Echocardiography.

**Table 2 genes-13-00320-t002:** Demographic, electrocardiographic, and echocardiographic characterization of healthy carriers and HCM patients.

		Healthy Carriers (*n* = 14)	HCM Patients (*n* = 25)	*p*
Demographic parameters				
	Age at first visit (mean)	25.1	51.4	<0.0001
	Age at last follow up visit (mean)	29.6	58.2	<0.0001
	Sex (% female)	49	46	ns
Electrocardiographic parameters				
	Left Ventricle Hypertrophy (% of patients)	0	30.4	0.033
	Pathologic Q waves (% of patients)	0	14.3	ns
	Pathologic Negative T waves (% of patients)	16.7	20	ns
Echocardiographic parameters				
	LVEF (%)	66.9	64.9	ns
	LV End Diastolic Diameter (mm)	43.8	45.3	ns
	LA Diameter (mm)	32.3	40.3	0.0012
	Maximum LV Wall Thickness (mm)	9	16.5	<0.0001
	RV Hypertrophy (% of patients)	0	4.8	ns
	Altered LV Diastolic Filling Pattern (% of patients)	8.3	27	ns

LA: Left Atrium, LV: Left Ventricle, LVEF: Left Ventricle Ejection Fraction, ns: not significant, RV: Right Ventricle.

**Table 3 genes-13-00320-t003:** Incidence of clinical events in patients with findings of the p.Arg652Lys variant in *MYH7*.

Clinical Events		
ACE (N-%)	5	20.0%
Sudden Death (N-%)	1	4.0%
SD under than 50 years of age (N-%)	0	0.0%
Aborted SD (N-%)	1	4.0%
Apropriate ICD discharge (N-%)	0	0.0%
HF (N-%)	2	8.0%
HF Admission (N-%)	0	0.0%
AF (N-%)	3	12.0%
Embolic Event (stroke or peripheral ischemic event) (N-%)	3	12.0%
SRT (N-%)	1	4.0%
Alcohol Ablation (N-%)	0	0.0%
Myectomy (N-%)	1	4.0%

ACE: Adverse Cardiovascular Event, AF: Atrial Fibrillation, ASD: Aborted Sudden Death, HF: Heart Failure, ICD: Implantable Cardiac Defibrillator, SD: Sudden Death, SRT: Septal Reduction Therapy.

**Table 4 genes-13-00320-t004:** Evidence framework for the pathogenicity classification of the p.Arg652Lys variant according to the criteria of the American College of Medical Genetics and Genomics (ACMG) and the Association for Molecular Pathology (AMP). In this diagram, the criteria are organised according to the type of evidence, as well as their strength.

ACMG/AMP Clasification.		
Class	Criteria for Interpretation	p.Arg652Lys Variant Interpretation [22]
Pathogenic Moderate (PM1)	Mutational hot spot and/or critical and well-established functional domain	UniProt protein MYH7_HUMAN domain ‘Myosin motor’ has 225 pathogenic variants out of 313 classified variants = 71.9% (greater than 66.7%).
Pathogenic Moderate (PM2)	Variant frequency and use of control populations	Variant not found in GnomAD exomes, with good GnomAD exomes coverage (87.1 is greater than 20.0).Variant not found in GnomAD genomes, with good GnomAD genomes coverage (31.3 is greater than 20.0).
Pathogenic Moderate (PM5)	Novel missense at the same position	Another amino acid missense variant at this position, Arg652Gly (chr14:23896451 T⇒C), is classified. Pathogenic, two stars (above the minimum of one star) by ClinVar and classified Likely Pathogenic using ACMG.
Pathogenic Supporting (PP2)	Variant spectrum	Missense variant in gene *MYH7* that has 412 pathogenic missense variants out of 417 pathogenic variants = 98.8%, which is greater than minimum of 66.7%, and only 21 benign missense variants. Gene *MYH7* is associated with Cardiomyopathy, dilated, 1S; Cardiomyopathy, familial hypertrophic; Myopathy, distal; and Myopathy, myosin storage, autosomal recessive.
Pathogenic Supporting (PP3)	Computational (in silico) data	Pathogenic computational verdict because of six pathogenic predictions from DANN, GERP, dbNSFP.FATHMM, MetaLR, MetaSVM, and MutationTaster (vs. three benign predictions from MutationAssessor, PROVEAN, and SIFT).

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
