# Peer review of "Genotype-Phenotype Correlation in Hypertrophic Cardiomyopathy: New Variant p.Arg652Lys in MYH7"

_genes, 2022, doi:10.3390/genes13020320_

Round 1

Reviewer 1 Report

The manuscript” Genotype-Phenotype correlation in Hypertrophic Cardiomyopathy: new mutation p.Arg652Lys in MYH7” by Guido Antoniutti elucidate the mutational spectrum of the MYH7 gene and genotype-phenotype correlations of HCM based on a small population of patients. The observation of this type would be highly significant even though the study is based on a small population of patients. However, the authors should summary and organize the results better to clearly delivery the information but not briefly list everything. HCM is more common in adults and many people are not diagnosed until adulthood. This is also observed in the cases with p.Arg652Lys in MYH7 as all the teenagers are healthy carriers.  Therefore, it should be considered to do the age-dependent analysis. Other minor Points: 1) provide more effective figure legends, such as it is confused for the definitions of “+/+”, “+” , “-”. 2) the author mentioned that the reason for studying p.Arg652Lys is it is not observed in the control. Please include the information of control, such as how many cases were examined, age range, gender information et al.

Author Response

Dear Reviewers,

We appreciate the work done in reviewing and suggesting changes that bring our article to the quality expected in the journal.

In response to you two I will list the changes made to the article with the means available to us:

  • We no longer say mutation. We rather refer to P, LP or VUS variants.
  • After reviewing all ACM criteria and the information available we have classified the p.Arg652Lys variant as Likely Pathogenic.
  • “Probably” pathogenic has been changed for Likely pathogenic.
  • We have included all 8 family trees as supplementary material and a reference of each family to the information in table 1 has been added.
  • We have reanalyzed the information on carriers and HCM patients, comparing demographic, electrocardiographic and echocardiographic characteristics, describing the main differences in the text and adding all relevant information at a new table (Table 2).
  • A complete age-dependent analysis on clinical events has been carried out but since there are just a few events, we believe it is best described subjectively. We have commented on the age of the events in the results, which is also available in table 1, and then analyzed their significance in the discussion.
  • We make a little comment on the possibility of a founder effect on our region. At our genetic lab are working hard to confirm that hypothesis but it takes time. Once it is finished we will make a new publication to communicate it.
  • Information has been better delivered with a few changes on the text and adding a new table.
  • Figure legends have been slightly modified for easier interpretation.
  • Since we mention that the variant is not present in controls, we have added the available information for the controls we used to check it, in public databases (GnomAD) and our private database.
  • Spelling mistakes marked have been corrected.

We hope to have achieved an article with the expected standards.

Sincerely.

Reviewer 2 Report

This is a remarkable and comprehensive genetic and clinical evaluation of individuals with hypertrophic cardiomyopathy from eight non-related families. The authors provide enough evidence to prove that c.1955G>A (p.Arg652Lys) variant in MYH7 leads to hypertrophic cardiomyopathy with reduced penetrance. The study should help in clinical diagnosis of individuals carrying this variant, and clinical management of probands and carrying family members.

Comment 1: Over the past few years, the genetic nomenclature “variant” is used in place of “mutation”. The intent is for “variant” to remain a neutral term. I would suggest the authors to replace the word “mutation” with “variant” all through the manuscript including the title.

Comment 2: Authors follow ACMG/AMP guidelines to classify c.1955G>A variant. They provide three pathogenic moderate (PM) and two pathogenic supporting (PP) criteria. According to ACMG/AMP guideline this would add up to ‘likely pathogenic’ not ‘pathogenic’. Authors should change the text accordingly unless they provide any pathogenic strong evidence, which should make the variant as pathogenic. Authors should check with clinical/medical geneticist to see if they could apply ‘PS4’ for their variant as the study involves multiple families.

Comment 3: The authors use the terminology “probably pathogenic”. However, ACMG guidelines (Richards et al.,) use the term “likely pathogenic”. Since authors have used ACMG criteria to classify the variant, they should consider using “likely pathogenic” instead of “probably pathogenic”.

Comment 4: Authors mention that they have gathered genealogical tree information of at least 3 generations, but they provide family chart for only two families (Figure 1 and 2). For the benefit of readers, the authors should consider including family tree for the remaining six families as supplemental data. In Figure 1 and 2, authors should add family number information that is corresponding to Table 1.

Comment 5: Authors have not done any further statistical analysis among the individuals who carry c.1955G>A variant. Authors should provide mean/ median age for individuals with or without HCM. Authors should also explore if it is possible to predict percent clinical manifestation of HCM at a particular age in individuals carrying this variant.

Comment 6: Authors mention about possible founder effect in the Balearic Islands, Spain as this variant has not been identified in any other population. Authors should consider collecting SNParray data for individuals carrying c.1955G>A variant to check for founder effect.

Comment 7: Spelling mistake in Table 1: Change ‘Follw-up time (months)’ to ‘Follow-up time (months)’.

Author Response

(The authors gave the same response as above.)
